# Verteporfin-Loaded Mesoporous Silica Nanoparticles’ Topical Applications Inhibit Mouse Melanoma Lymphangiogenesis and Micrometastasis In Vivo

**DOI:** 10.3390/ijms222413443

**Published:** 2021-12-14

**Authors:** Nausicaa Clemente, Ivana Miletto, Enrica Gianotti, Maurizio Sabbatini, Marco Invernizzi, Leonardo Marchese, Umberto Dianzani, Filippo Renò

**Affiliations:** 1Department of Health Science and Interdisciplinary Research Center of Autoimmune Disease (IRCAD), Università del Piemonte Orientale, Via Solaroli, 17, 28100 Novara, Italy; nausicaa.clemente@med.uniupo.it (N.C.); umberto.dianzani@med.uniupo.it (U.D.); 2Department of Science and Technology Innovation, Università del Piemonte Orientale, Via T. Michel, 11, 15121 Alessandria, Italy; ivana.miletto@uniupo.it (I.M.); enrica.gianotti@uniupo.it (E.G.); maurizio.sabbatini@uniupo.it (M.S.); leonardo.marchese@uniupo.it (L.M.); 3Department of Health Science, Physical Medicine and Rehabilitation, Università del Piemonte Orientale, Via Solaroli, 17, 28100 Novara, Italy; marco.invernizzi@med.uniupo.it; 4Department of Integrated Research and Innovation, Translational Medicine (DAIRI), Hospital S.S. Antonio e Biagio e Cesare Arrigo, 15121 Alessandria, Italy; 5Department of Health Sciences and Innovative Research Laboratory for Wound Healing, Università del Piemonte Orientale, Via Solaroli, 17, 28100 Novara, Italy

**Keywords:** melanoma, lymphoangiogenesis, micrometastasis, photodynamic therapy, verteporfin, mesoporous silica nanoparticles, B16-F10 cells

## Abstract

Photodynamic therapy (PDT) has been pointed out as a candidate for improving melanoma treatment. Nanotechnology application in PDT has increased its efficacy by reducing side effects. Herein, mesoporous silica nanoparticles (MSNs) conjugated with verteporfin (Ver-MSNs), in use with PDT, were administered in mice to evaluate their efficacy on lymphoangiogenesis and micrometastasis in melanoma. Melanoma was induced in mice by the subcutaneous injection of B16-F10 cells. The mice were transcutaneously treated with MSNs, Ver-MSNs, or glycerol and exposed to red light. The treatment was carried out four times until day 20. Lymphangiogenesis and micrometastasis were identified by the immunohistochemical method. Lymphoangiogenesis was halved by MSN treatment compared with the control animals, whereas the Ver-MSN treatment almost abolished it. A similar reduction was also observed in lung micrometastasis. PDT with topically administrated Ver-MSNs reduced melanoma lymphoangiogenesis and lung micrometastasis, as well as tumor mass and angiogenesis, and therefore their use could be an innovative and useful tool in melanoma clinical therapy.

## 1. Introduction

Cancer spread or metastasis is the most important predictor of patient prognosis, so the main effort to address a cancer form is to study and counter the spread of cancer cells [1]. It is widely known that most cancers spread their metastases through the lymphatic vessels that drain liquids and cells from the tissue microenvironment. Therefore, the blind-bottomed lymphatic capillaries, given their high permeability, give access to metastatic cells [2].

Clinical studies have indicated that lymphatic vessels may be colonized by hosting small nodules of tumoral cells, having the potential to disseminate secondary tumors over a long period of time [3,4]. This may give the lymphatic vessels a critical role in the tumor dissemination process.

Cutaneous melanoma (CM) is a highly metastatic tumor characterized by rapid systemic dissemination [5]. Malignant melanoma adopts a particular system of invasion of the organs through the lymphatic vessels. One active melanoma dissemination strategy is lymphangiogenesis (the formation of lymphatic vessels associated with tumor growth) [6,7], as confirmed by the work of Lund et al. [8], where spontaneous lung metastasis from primary melanomas was decreased in a mouse strain lacking dermal lymphatic vessels.

The highly metastatic tendency of CM makes it particularly difficult to be treated with radiotherapy and chemotherapy, which are unable to effectively counter the development of highly diffusive metastases [9]. Therefore, alternative therapies such as photodynamic therapy (PDT), which was recently approved by the Food and Drug Administration (FDA), are also used. PDT combines a photosensitizer, administered or applied locally to the target, and a specific wavelength light capable of penetrating tissues. When the photosensitizer is irradiated by adequate light, it generates reactive oxygen species (ROS) [10] which kill the malignant cells, cause the destruction of the vascular structure, and lead to activation of the inflammatory response [11]. Furthermore, in order to improve the bioavailability of drugs, to protect them from degradation, and to increase their penetration by specifically targeting cancer cells or the tumor environment [12], different types of nanocarriers have been used [13,14,15].

Verteporfin (Ver), a hydrophobic benzoporphyrin derivative, is a photosensitizer largely employed in PTD [16,17,18]. Our group already used Ver conjugated with mesoporous silica nanoparticles (MSNs) [17,19] (Figure 1) to topically treat mouse melanoma obtaining both tumor angiogenesis and mass reduction [20].

In this part of an in vivo study, we evaluated the efficacy of photodynamic therapy with verteporfin-loaded mesoporous silica nanoparticles to reduce tumor lymphoangiogenesis and affect micrometastasis.

## 2. Results

The efficacy of Ver-MSNs used in photodynamic therapy was tested in vivo in an animal model of melanoma. When the tumors became palpable, the mice were treated transcutaneously with glycerol, MSNs, and Ver-MSNs and then irradiated with a red light. Treatment was repeated four times every 4 days. At the end of treatment, the Ver-MSN-treated animals displayed significantly lower tumor growth and angiogenesis than those treated with either glycerol or MSNs. Tumor lymphoangiogenesis was evaluated in these samples by Lymphatic Vessel Endothelial Receptor 1 (LYVE-1) immunostaining (Figure 2). LYVE-1 is considered a lymphatic specific marker, and it is a CD44 homolog found primarily on lymphatic endothelial cells. As shown in Figure 2, LYVE-1 fluorescence was widespread throughout the tumor sections in the control samples and occupied 25.2 ± 1.5% of the investigated area (0.96 mm^2^), whereas treatment with MSNs reduced the fluorescence area by 14.2 ± 0.8%, where the LYVE-1-positive cells were visualized in discrete portions of the sample. Ver-MSN treatment was able to almost turn off the LYVE-1 signal (fluorescence present in 0.003 ± 0.0001% of the survey area), indicating a complete abolishment of lymphangiogenesis.

A number of studies have indicated that increased lymphatic vessel invasion significantly enhances the risk of lymph node metastasis, distant metastasis, and death [21]. In order to correlate lymphoangiogenesis with the melanoma dissemination in our animal model, the lungs, livers, kidneys, and spleens of mice were collected and examined with an antibody cocktail of anti-melanosome (HMB45), anti-MART-1/melan A (A103), and anti-tyrosinase (T311) mouse monoclonal antibodies. However, only the lung samples showed the presence of micrometastasis, whereas no clear micrometastasis was scored in other organs (data not shown). Significantly, as shown in Figure 3A, the untreated tumor-bearing animals’ lungs showed the presence of micrometastasis. The average number ± standard error (S.E.) of micrometastasis scored in 10 different lung slices in the control animals was 22.70 ± 1.48 (Figure 3B), whereas the topical treatment with MSNs significantly reduced the micrometastasis number (6.81 ± 0.53, Figure 3A,B). Finally, lung micrometastasis was almost abolished by Ver-MSN treatment (2.33 ± 0.49, Figure 3A,B).

## 3. Discussion

Malignant melanomas, as well as many other human cancers, first metastasize via lymphatic vessels to the sentinel lymph node through recently clarified mechanisms. A number of studies revealed that tumor-induced lymphangiogenesis plays an important and active role in the promotion of cancer metastasis to the lymph nodes [22]. This is also because lymphatic vessels, which have high permeability and a lack of tight junction structure compared with blood vessels, are particularly accessible to tumor cells [23]. Furthermore, the tumor micro-environment presents a high interstitial pressure that may offer the push to make the tumor able to enter lymphatic vessels [24].

Lymphangiogenesis is a largely absent process under normal physiological postnatal conditions, taking place during pathological conditions (e.g., inflammation, tissue repair, and tumor growth) [21]. Tumor-induced lymphangiogenesis is induced by lymphangiogenic growth factors secreted by tumor cells, stromal cells, and tumor-infiltrating macrophages [21]. The interplay between tumorigenesis and lymphangiogenesis is now considered a key target to modulate metastatic spread [7]. Typically, targeting the lymphoangiogenic pathway regulated by Vascular Endothelial Growth Factor (VEGF) C or D using antibodies to neutralize lymphatic growth factors or receptors, or a soluble form of VEGFR-3 to trap VEGF-C/D, appears to reduce or suppress the rate of lymph node metastasis in mouse models [7].

When we topically treated induced melanoma in mice with verteporfin-grafted mesoporous silica nanoparticles (Ver-MSNs) and irradiated the tumor using a red light, we observed a reduction in intratumor lymphatic vessels.

MSNs alone reduced lymphoangiogenesis, even if they did not release single oxygen species [20]. We can hypothesize that MSNs induced a direct toxicity in lymphoendothelial cells because these nanoparticles could be engulfed but not metabolized, causing cell death, as previously observed [25].

On the other hand, the Ver-MSN-induced reduction in lymphatic vessels could be due to either direct or indirect effects. In particular, we observed that in this animal model, Ver-MSNs reduced the tumor mass [20]. Therefore, a small tumor could secrete a low quantity of lymphoangiogenic factors. Conversely, it has already been shown by both in vivo and in vitro studies that PDT using verteporfin affected lymphoangiogenesis, inducing both apoptosis and autophagy [26]. It is also noteworthy that PDT seemed to affect lymphoendothelial cells selectively [26]. Reduced lymphangiogenesis resulted in less micrometastasis observed in the animals’ lungs. We did not observe micrometastasis in the kidneys or any other organ, probably because the B16-F10 melanoma cells used in our animal model bound selectively to 90 kD lung endothelial cell adhesion molecule-1 (Lu-ECAM-1) [27], inducing lung metastasis.

## 4. Materials and Methods

### 4.1. Synthesis of Verteporfin-Loaded Mesoporous Silica Nanoparticles

Verteporfin-loaded mesoporous silica nanoparticles were prepared according to the literature’s procedures [17,19]. All the reagents used were purchased from Sigma-Aldrich (Milano, Italy). Brefly, the MSNs were synthesized using cetyltrimethylammonium bromide (CTAB) and tetraethylorthosilicate (TEOS). CTAB was dissolved in distilled water, NaOH and TEOS were added, and the mixture was stirred at 80 °C for 2 h. The white precipitate formed was filtered off and washed, and the CTAB surfactant was removed. The dried MSNs were suspended in toluene, aminopropyltriethoxysilane (APTES) was added, and the mixture was stirred overnight to obtain aminopropyl-functionalized mesoporous silica nanoparticles (amino-MSNs). The amino-MSNs were suspended in dimethylformamide (DMF) containing verteporfin (Ver), 1-[Bis(dimethylamine)methylene]-1H-1,2,3-triazolo[4,5-b] pyridinium3-oxide hexafluoro-phosphate (HATU, 1 eq.), and N,N-diisopropylethylamine (DIPEA, 1 eq.) to form Ver-based silica nanoparticles. Ver-MSNs were prepared with a nominal Ver loading of 40 mg/g. This loading value was chosen based on previous works [19,20].

### 4.2. In Vivo Melanoma Topical Treatment

Female 8-week-old C57BL/6J mice (The Jackson Laboratory, Bar Harbor, ME, USA) were bred under pathogen-free conditions in the animal facility of the Università del Piemonte Orientale and treated in accordance with the University Ethical Committee and European guidelines. The mice were injected subcutaneously with B16-F10 cells (2.5 × 10^5^ in 100 μL/mouse), and the tumor growth was monitored every 4 days. Eleven days after tumor induction, when the tumor was palpable, the mice were treated via transcutaneous administration of 2 mL of a solution of MSNs (5 µg/mL) in glycerol, Ver-MSNs (5 µg/mL) in glycerol, or the same volume of glycerol as a negative control (control animals (CTR)) [20]. Five minutes after each transcutaneous administration, the mice were exposed to a red light for 10 min. The treatment was carried out 4 times every 4 days, and the mice were sacrificed 4 days after the last administration or when they displayed sufferance. Five animals were employed for each group.

### 4.3. Immunofluorescence and Immunohistochemistry of Tumors and Animal Specimens

After euthanasia, all animals underwent complete necropsy. The tumor samples were embedded in optimal cutting temperature (OCT) compound (Bioptica SpA, Milano, Italy, snap-frozen, and stored at −80 °C until use. The tumor tissues were cut with a cryostat (thickness 5–6 µm) and treated with 4% paraformaldehyde (Sigma-Aldrich) diluted in PBS for 5 min at room temperature to fix the sample on the glass slides. The samples were then blocked with 5% Normal Goat Serum (R&D System, Minneapolis, USA) in PBS for 1 h to block aspecific sites. To detect lymphatic endothelial vessels, the slides were incubated with the primary polyclonal antibody anti-LYVE1 (1:50, Abcam, Cambridge, UK) overnight at 4 °C. The antibody was detected using an anti-rabbit Ig Alexa fluor 488-conjugation (1:400; Thermo Fisher, Waltham, MA, USA). Then, the sections were counterstained with 0.5 mg/mL of the fluorescent dye 4,6-diamidino-2-phenylindole-dihydrochloride (DAPI, Sigma-Aldrich) for 5 min to color the cell nuclei and then mounted using Prolong anti-fade mounting medium (Slow Fade AntiFADE Kit, Molecular Probes Invitrogen, Eugene, OR, USA). The complete procedure was performed in a humidified chamber. The sections were then observed by a fluorescence microscope (Leica, Italy) and analyzed with the Image Pro Plus Software for micro-imaging 5.0 (Media Cybernetics, version 5.0, Bethesda, MD, USA). The lungs, livers, kidneys and spleens were collected and stored in 10% neutral buffered formalin for histological evaluation of micrometastasis. The samples were paraffinized, sectioned (5-µm thick), stained with hematoxylin/eosin or Melanoma Triple Cocktail immunohistochemistry and evaluated by light microscopy. The primary antibodies included anti-melanosome (HMB45), anti-MART-1/melan A (A103), and anti-tyrosinase (T311) mouse monoclonal antibodies. This antibody cocktail demonstrated cytoplasmic staining and was used to aid in the identification of melanoma. Antibodies were detected using the avidin–biotin–peroxidase complex technique with the Vectastain ABC-AP Kit (Vector Laboratories, CA, USA). The reaction was visualized using the chromogen VECTOR^®^ SG Peroxidase (HRP) Substrate Kit (Vector Laboratories), which produces blue-gray reaction products. The nuclei were counterstained with hematoxylin and eosin staining. Positive and negative immunohistochemistry controls were routinely used. All reactions were visualized by light microscopy and assessed blindly by two observers, and the discordant cases were reviewed at a multi-head microscope until a consensus was reached. Each slide for histological staining was captured with a Nikon DS-Fi1 digital camera (Nikon, Shinjuku, Japan) coupled to a Zeiss Axiophot microscope (Zeiss, Oberkochen, Germany) using a 40× objective lens. NIS-Elements F software (V4.30.01, Nikon, Shinjuku, Japan) was used for image capturing. Each immunohistochemical marker was evaluated in at least 10 different slides (Image Pro Plus analysis system—Media Cybernetics, version 5.0, Bethesda, MD, USA).

### 4.4. Statistical Analysis

An ANOVA test followed by Bonferroni’s post hoc test was used for statistical analysis. Statistical procedures were performed with the GraphPad Instat statistical software (GraphPad Software Inc., CA, USA). Probability values of *p* < 0.05 were considered statistically significant.

## 5. Conclusions

In summation, we evaluated the efficacy of photodynamic therapy with verteporfin-loaded mesoporous silica nanoparticles in an in vivo model. The results showed that red light-irradiated Ver-MSNs were able to significantly reduce lymphatic invasion and micrometastasis tumor growth. Considering the importance that lymphangiogenesis has in melanoma metastasis, our work further confirms verteporfin mesoporous silica nanoparticles as a valuable tool to treat cutaneous melanoma.

## Figures and Tables

**Figure 1 ijms-22-13443-f001:**
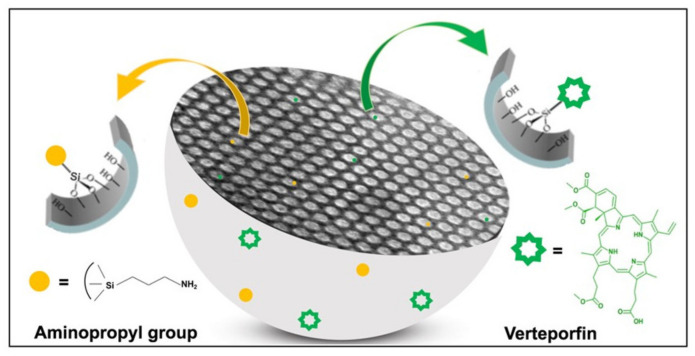
Pictorial representation of amino-modified mesoporous silica nanoparticles grafted with Ver.

**Figure 2 ijms-22-13443-f002:**
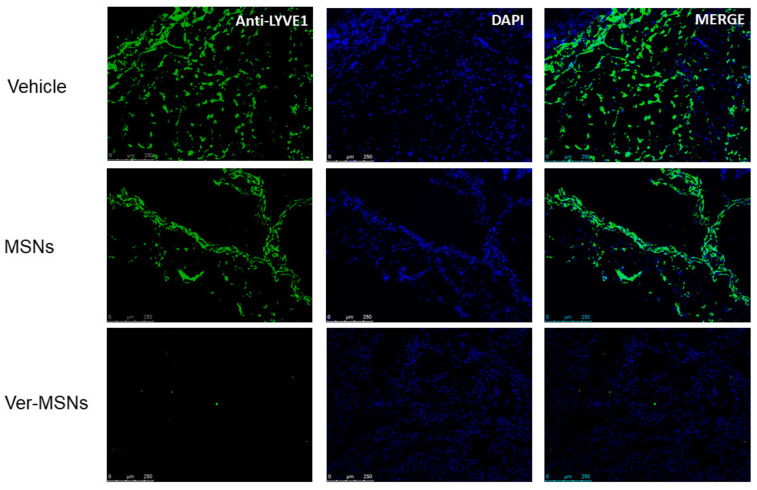
Immunofluorescence anti-LYVE1 on melanoma tumor tissues. Mice with palpable subcutaneous tumors were treated every 4 days with glycerol (Vehicle), MSNs, or Ver-MSNs, with representative images of B16-F10 melanoma stained for LYVE1 (green) and, to view the nucleus, with DAPI (blue). Lymphatic vessels occurring in tumor mass of the vehicle group were reduced by MSN, while treatment with Ver-MSNs was able to abolish the anti-LYVE1 immunofluorescence. Bar = 250 µm.

**Figure 3 ijms-22-13443-f003:**
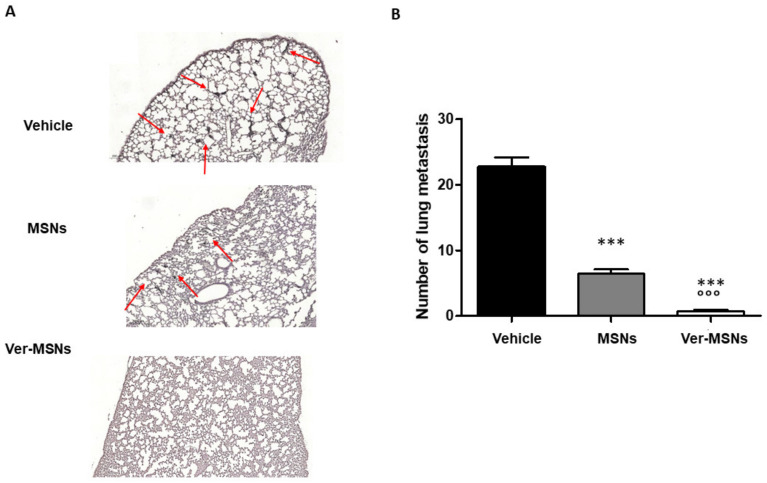
Effect of Ver-MSNs on the metastasizing ability to colonize the lung of B16-F10 tumors in C57BL/6J mice. (**A**) Immunohistochemical staining on lung anti-Triple M. Positive metastasizing nodules in the lung parenchyma are indicated by arrows. Bar = 200 µm. (**B**) Quantification of metastasis numbers counted in at least 10 different fields. The high number of metastasizing mass occurring in control lungs is strongly reduced following the MSN treatment and almost abolished following Ver-MSN treatment. A one-way ANOVA test and Dunnett’s multiple comparisons test were used to compare differences in the treatments. *** *p* < 0.0001 vs. vehicle. °°° *p* < 0.0001 vs. MSNs.

## Data Availability

Data will be available following direct request to corresponding author.

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
