# Peer review of "Verteporfin-Loaded Mesoporous Silica Nanoparticles’ Topical Applications Inhibit Mouse Melanoma Lymphangiogenesis and Micrometastasis In Vivo"

_ijms, 2021, doi:10.3390/ijms222413443_

Round 1

Reviewer 1 Report

Manuscript by Clemente et al. has studied the effect of verteporfin-loaded mesoporous nanoparticles in use with PDT to mouse melanoma lymphoangiogenesis and micrometastasis in vivo. The authors have nicely shown that PDT with topically administrated Ver-MSNs reduced melanoma lymphoangiogenesis and lung micrometastasis and therefore could be used as an innovative tool in melanoma clinical therapy.

The question is very topical and results promising. However, I have few questions to specify some issues.

  1. The discussion section is poorly written. Please discuss why MSNs alone (together with light) has such a strong effect on lymphoangiogenesis and micrometastasis in vivo.
  2. In your previous study (Clemente et al. 2019 J. of Photochemistry and Photobiology) treatment with MSNs enhanced tumor angiogenesis (by staining with CD31), however reduced melanoma growth in mice. Can you please discuss your current results in the light of previous study?
  3. What is the effect of MSNs and Ver-MSNs without irradiation with a red light?

Minor points.

  1. In Conclusion section, the first sentence (lines 223-224) is not relevant. The second sentence (To sum up, …; 224-225) is a copy-paste from the previous paper and is not accurate – you have not used in vitro model in the current work.

Author Response

Review #1

Major point

The discussion section is poorly written. Please discuss why MSNs alone (together with light) has such a strong effect on lymphoangiogenesis and micrometastasis in vivo.

In the discussion section we added: “MSNs alone reduced lymphoangiogenesis even if they do not release single oxygen [17]. We can hypothesize that MSNs induced a direct toxicity in lymphoendothelial cells because these nanoparticles can beengulfed, but not metabolized causing cell death as previously observed [25].

In your previous study (Clemente et al. 2019 J. of Photochemistry and Photobiology) treatment with MSNs enhanced tumor angiogenesis (by staining with CD31), however reduced melanoma growth in mice. Can you please discuss your current results in the light of previous study?

The authors apologize for the mistake that has not been corrected in article cited. In fig.6a from the cited article (Clemente et al.2019), the angiogenesis increased in the presence of MSNs but the increase was not statistically significant as indicated (please check the standard deviation bar).

What is the effect of MSNs and Ver-MSNs without irradiation with a red light?

In the in vivo experiments MSNs and Ver-MSNs effects were evaluated only after red light exposure, as their effects in the absence of red light was already tested in vitro, showing that Ver-MSNs were able to induce a small toxic effects also in the dark (Clemente et al. 2019 J. of Photochemistry and Photobiology).

Minor points:

Sentence in lines 223-224 has been eliminated.

Sentence  in lines 224-225 has been changed.

Reviewer 2 Report

This is a simple but important experiment design. The authors used an in vivo mouse model to induce melanoma, than by subcutaneous injection of Mesoporous-silica-nanoparticles conjugated with Verteporfin (a as photodynamic therapy)  was used to eliminate cancer cells.

Notes: abbreviations for the same terms are used several times

The description of MSNs is not consistent: Mesoporous Silica Nanoparticles; mesoporous silica nanoparticles; Mesoporous-silica-nanoparticles.

How many images were taken after the treatment, to evaluate effect of MSBs?

What is the vehicle? What is CTR?

There are two Figure 2 are those connected? There is no Figure 3.

The descriptions of Figure legends are not consistent.

The discussion is overall.

The references are not up-to date; the only new reference is from 2020.

Author Response

Reviewer #2

Notes: abbreviations for the same terms are used several times

All the abbreviations have been carefully checked.

How many images were taken after the treatment, to evaluate effect of MSBs?

As indicated in M&M (end of par 4.3): “Each immunohistochemical marker was evaluated in at least ten different slides”.

What is the vehicle? What is CTR?

As indicated in M&M (par 4.2) : “mice were treated via transcutaneous administration of 2 ml of solution of MSNs (5 µg/ml) in glycerol, Ver-MSNs (5 µg/ml) in glycerol or the same volume of glycerol as negative control”.

There are two Figure 2 are those connected? There is no Figure3

Authors apologise for the typos. There are three different images: Fig.1 for MSNs structure, Fig.2 for linfoangiogenesis and Fig,.3 for metastasis.

The descriptions of Figure legends are not consistent.

Authors disagree with Reviewer #2

The discussion is overall.

Discussion has been improved also following Reviewer #1 suggestions.

The references are not up-to date; the only new reference is from 2020.

Three more updated references about nanoparticles use in melanoma treatment have been added:

Zhang et al.; 2020

Battaglia et al., 2021

Gao et al.; 2021

Round 2

Reviewer 1 Report

Thank you for clarifying these issues.

Author Response

The authors thank the reviewer for suggestions. 

Reviewer 2 Report

Abstract

Ver-MSNs abbreviation is missing

Introduction:

Lines 48-6-48: and 52-54 This two sentences are similar, it should be combined.

Lines 52-54: the reference 3 is cited 2 times within one sentence, it would be enough at the and of the sentence.

Line 57: Lund and coll. change to Lund et al.

Line 70: Change [13,14,15] to [13-15].

Line 71: Verteporfin (Ver) abbreviation was already used

Line 72: PTD [14,1516,17,18], use PTD [16-18].

Figure 2 legend: The magnification is missing

Figure 3 legend: The description of A. and B. is missing from the legend.

Discussion:

Line 148: typing mistake " handm "

Materials and Methods:

Abbreviations should be used only once

Lines 160 and 161; Line 170: Verteporfin (Ver),

Line 174: the number of references (16,17) are not correct

Line 183: What is (CTR)? Reference [17] is it correct?

Line 201: Molecular Probes Invitrogen... country is missing

Lines 212 and 213 ..Vector Laboratories... country missing

The authors disagree with Reviewer #2: I disagree with the authors: The descriptions of Figure legends are still not consistent.

The authors inserted new references, but the references numbers are not always correct

Author Response

Review #2

Abstract

Ver-MSNs abbreviation is missing

Abbreviation added

Introduction:

Lines 48-6-48: and 52-54 This two sentences are similar, it should be combined.

Sentences have been rearranged.

Lines 52-54: the reference 3 is cited 2 times within one sentence, it would be enough at the and of the sentence.

Corrected

Line 57: Lund and coll. change to Lund et al.

Corrected

Line 70: Change [13,14,15] to [13-15].

Corrected

Line 71: Verteporfin (Ver) abbreviation was already used

Corrected

Line 72: PTD [14,1516,17,18], use PTD [16-18].

Corrected

Figure 2 legend: The magnification is missing

In fig.2 there is a bar on the left bottom part of the figure. Information has been added to the legend.

Figure 3 legend: The description of A. and B. is missing from the legend.

Description for A and B have bee added

Discussion:

Line 148: typing mistake " handm "

Corrected

Materials and Methods:

Abbreviations should be used only once

Lines 160 and 161; Line 170: Verteporfin (Ver),

Corrected

Line 174: the number of references (16,17) are not correct

References have been corrected. Now ref 19 and 20.

Line 183: What is (CTR)? Reference [17] is it correct?

CTR=control samples, animals treated only with glycerol. Reference 17 has been corrected (now ref 20)

Line 201: Molecular Probes Invitrogen... country is missing

Lines 212 and 213 ..Vector Laboratories... country missing

Country added

The authors disagree with Reviewer #2: I disagree with the authors: The descriptions of Figure legends are still not consistent.

Authors tried to make legends more consistent

The authors inserted new references, but the references numbers are not always correct

All the references numbers have been corrected